# The Practice and Potential of Renewable Energy Localisation: Results from a UK Field Trial

**Peter Boait** [1,*] **, J. Richard Snape** [1] **, Robin Morris** [2] **, Jo Hamilton** [3] **and Sarah Darby** [4]

1   Institute of Energy and Sustainable Development, De Montfort University, Queens Building, The Gateway, Leicester LE1 9BH, UK; jsnape@dmu.ac.uk

2   Energy Local (Development) Ltd., Crickhowell Resource and Information Centre, Beaufort Street, Crickhowell, Powys NP8 1BN, UK; robin@energylocal.co.uk

3   Department of Geography and Environmental Science, School of Archaeology, Geography and Environmental Science, The University of Reading, Whiteknights, P.O. Box 227, Reading RG6 6AB, UK; e.j.hamilton@pgr.reading.ac.uk

4   Environmental Change Institute, School of Geography and the Environment, University of Oxford, South Parks Road, Oxford OX1 3QY, UK; sarah.darby@ouce.ox.ac.uk

*   Correspondence: p.boait@dmu.ac.uk; Tel.: +44-774-064-4211

**Abstract:** The adaptation of electricity demand to match the non-despatchable nature of renewable generation is one of the key challenges of the energy transition. We describe a UK field trial in 48 homes of an approach to this problem aimed at directly matching local supply and demand. This combined a community-based business model with social engagement and demand response technology employing both thermal and electrical energy storage. A proportion of these homes (14) were equipped with rooftop photovoltaics (PV) amounting to a total of 45 kWp; the business model enabled the remaining 34 homes to consume the electricity exported from the PV-equipped dwellings at a favourably low tariff in the context of a time-of-day tariff scheme. We report on the useful financial return achieved by all participants, their overall experience of the trial, and the proportion of local generation consumed locally. The energy storage devices were controlled, with user oversight, to respond automatically to signals indicating the availability of low cost electricity either from the photovoltaics or the time of day grid tariff. A substantial response was observed in the resulting demand profile from these controls, less so from demand scheduling methods which required regular user configuration. Finally results are reported from a follow-up fully commercial implementation of the concept showing the viability of the business model. We conclude that the sustainability of the transition to renewable energy can be strengthened with a community-oriented approach as demonstrated in the trial that supports users through technological change and improves return on investment by matching local generation and consumption.

**Keywords:** community energy; energy storage; time of use tariff; home battery; demand response; renewable energy; business model

## 1. Introduction

Community energy initiatives are widely recognised as a valid and useful response to the sustainability challenges of climate change, energy security, and energy affordability. The UK government published a Community Energy Strategy in 2014 [1], updated 2015 [2], aimed at encouraging both supply and demand side projects. Municipal and co-operative ownership models providing renewable generation capacity are playing a major role in Germany's "Energiewende" [3], while the USA's 900 rural electricity co-operatives [4], founded as a response to economic depression in the 1930s, are evolving to promote energy efficiency and adopt low carbon generation. Community

energy schemes can have many organizational forms based on community of place or interest, but surveys and reviews such as [5–8] identify as typical benefits their ability to engage consumer participation in the systemic changes taking place and a contribution to societal cohesion through shared goals and a fair and transparent allocation of financial costs and returns. Another benefit found by [9] is that community members have a more favourable attitude overall to local renewable energy installations, mitigating the "not in my backyard" attitude that is otherwise common. However these studies also indicate that, at least in the UK, schemes are often financially fragile and depend on enthusiastic volunteers, so need policy and regulatory support to flourish.

An opportunity to strengthen the economic basis for community energy arises from the rising demand for electricity expected as electrification of transport and heating takes place. This prospect was reinforced in the UK by the government's publication of a policy to ban the sale of most petrol and diesel vehicles by 2040 [10], a goal also set by the government of France [11]. The UK's electricity system operator, National Grid, predicted in their 2017 Future Energy Scenarios report [12] that with consumer engagement in end-use energy efficiency and demand response for system efficiency, peak demand by 2050 is limited to 74 GW (from a baseline of 62 GW), but rises to 85 GW without it. So there should be scope for rewarding communities for their contribution to a more efficient solution, which earlier studies such as [13] have valued as worth up to £30 Bn in deferred or avoided network reinforcement costs. Reflecting this potential, the 2018 Future Energy Scenarios report from National Grid [14] includes "Community Renewables" as one of the scenarios that can deliver UK commitments to the Paris Agreement.

The community contribution can be seen as the product of service expectations, activities and technologies within a given community at a given time: a 'demand response space' [15]. There may well be greater opportunities for developing demand response at community scale rather than focusing on individual customers. These opportunities could arise from norms and practices developed through social learning about new technologies and processes [16], from trust-building [17]; and from the diversity of activities, skills and technologies to be found within communities [18].

In this paper we report the results from trialing a combination of business model and "smart home" technology designed for communities of place whose common factor is that they reside on the same segment of the local electricity distribution network—e.g., they might share the same low voltage (LV) network. An assumption of the model is that there is some distributed low carbon electricity generation on the shared LV network. This can take any of the common forms such as solar photovoltaics (PV), wind generation, combined heat and power (CHP) or micro hydro. The technology and the incentives from the business model are then framed to work synergistically with community activities to empower participants to reduce their cost of electricity through three mechanisms in order of priority:

- adapting demand to make use of the local generation wherever possible;
- avoiding use of non-local electricity at high cost times such as early evening;
- reducing overall consumption of electricity.

A benefit of the proposed business model is that it overcomes a legal constraint on financing local generation by forming a community co-operative. UK financial regulation requires that the investing members of a co-operative must either be workers in, or consumers of, the commercial product of the enterprise. This is to avoid the regulatory concessions available to co-operatives being exploited by purely speculative investment offers. Where the whole output of a generator is sold to an electricity supplier through a power purchase agreement, under a recent UK regulatory clarification [19] the investors in the generator cannot be considered consumers. But as this model allows consumers to purchase locally-generated electricity they can form a co-operative to fund the generator, which has advantages over other forms of legal entity such as a Community Interest Company (CIC) in allowing more flexibility in the use of profits.

There have been many trials of time-of-day tariffs and use of technology to influence patterns of residential electricity consumption, as for example summarised in [20,21]. More recently the potential of energy storage to contribute to demand flexibility has been recognized [22]. Drawing on lessons from that experience, the present trial incorporated a comprehensive combination of features and demand response measures not previously tested in the UK. These were:

- a time-of-use tariff with a static baseline and a day-ahead dynamic adjustment reflecting the predicted availability of locally-generated electricity from PV panels owned by some participants;
- a web-based display of the current tariff and consumption on user's smart phones, tablets, and computers;
- technology to automatically schedule loads at an optimum time with respect to the tariff while prioritizing user needs and preferences;
- exploitation of domestic energy storage in batteries and thermal storage heaters;
- regular feedback on the financial savings achieved by individual users and the participant group as a whole;
- a sustained program of engagement aimed at retaining user interest and obtaining their feedback.

The goals of the project (called CEGADS) undertaking this trial were to demonstrate the viability of the business model, test the acceptability of this level of innovation to consumers, and evaluate the amount of demand-side response to the measures deployed. The remainder of the paper is structured as follows. In Section 2 we describe the participant community, the business model, and the technology employed. Section 3 provides the results obtained, in respect of the use of local generation, financial outcome, demand side response, and the experience of participants. Section 4 describes briefly an agent-based modelling study of the scheme implemented, and a follow-up project implementing the business model now in fully commercial operation. The overall implications of the findings are discussed in Section 5 followed by conclusions.

## 2. The CEGADS Trial

### 2.1. The Participants and Business Model

The project name CEGADS stands for Community Electricity Generation, Aggregation, and Demand Shaping, indicating the key features involved. The project is also known by the acronym SWELL, referring to the Energy Local model in the cluster of Oxfordshire villages Shrivenham, Watchfield, and Longcot, in which the trial took place and the 48 participating households were recruited. Many of the residents had already subscribed to the local Westmill energy co-operatives [23,24] operating substantial wind and solar farms, but these generators were subject to wholesale power purchase agreements (entered into prior to the regulatory clarification at [19] summarised above), and being connected at 33 kV were not accessible to the present scheme. However the charitable trust associated with these co-operatives was thereby able to facilitate recruitment for this project from an informed community. Metering of electricity consumption and generation at one-minute intervals was installed in each household, along with the display and control technology. This equipment was installed at no cost to users, as were the batteries described later.

The generation for CEGADS was provided by roof-mounted PV panels already owned by 14 participants with a total capacity of 45 kWp. The export electricity from these panels (i.e., the generated electricity not consumed within the household) was metered and aggregated to form a resource which was considered available for supply at a favourable tariff (£0.065/kWh) to the remaining non-generating participants. The allocation of this export energy aggregate $A$ in each half-hour was computed by finding iteratively a "fill level" $L$ such that for each of $n$ consumers with demand $e_i$ in the half hour greater than $L$, $L$ kWh would be considered supplied from $A$, and for those

remaining *m* consumers with demand $e_j$ less than *L*, their demand would be fully met from *A*, with *L* also satisfying:

$$A = nL + \sum_{j=1}^{j=m} e_j \tag{1}$$

where *A* was large enough to more than supply all non-generating consumers then the residue was considered community export. This method of fair allocation of local generation is a key feature of the business model, which also allows the community export to be sold to an electricity supplier under a power purchase agreement. To enable participants in the trial to retain their existing electricity supplier and tariffs, the time-of-use tariff applied to electricity not generated locally was implemented as an incentive scheme where the difference between the actual cost of electricity to participants and the cost they would have incurred under the trial tariff is given to them in the form of credit vouchers exchangeable for goods at a supermarket chain associated with the electricity supplier supporting the project. The commercially-realistic (for 2016) time of use tariff rates offered are shown in Figure 1. Generators were credited with £0.065 for each export kWh matched with consumption, and £0.055 for each kWh not matched so considered as taken up by a power purchase agreement (PPA).

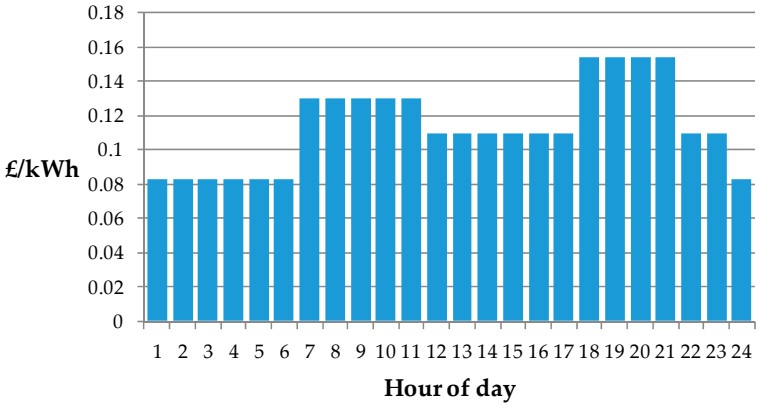

**Figure 1.** Time-of-use electricity tariff rates.

### 2.2. The Metering and Demand Response System

To execute the metering required for this scheme and enable the participants to make best use of the local generation and time-of-use tariffs a smart metering and control unit was installed in each household. Branded "Hestia", this unit provided from an internal web server a display of the tariff rates on any convenient IT device connected to the household broadband, but modified with a dip in the displayed rates during the middle of the day that reflected approximately the amount of local PV generation predicted to be available based on the overnight local weather forecast. It also provided displays of electricity consumption and generation over the last 24 h for the household, the participant community as a whole, the aggregate PV generation, and the PV generation for the household for those so equipped. To provide the data for these displays, metering data at one minute intervals was collected and processed in a central database using a commercial cloud service. A simplified view of the system is shown in Figure 2.

The Hestia control unit also performed automatic demand response for controllable appliances as illustrated in Figure 2. Six of the participating dwellings had space heating provided by electrically-heated thermal storage heaters and hot water from an immersion-heated tank. In aggregate these appliances provided about 60 kWh of thermal storage in each of the six homes. Charging of these useful thermal energy stores was controlled such that user comfort requirements as expressed on the Hestia user interface were prioritized, but was otherwise optimized against a tariff-dependent signal from the database server that ensured cost effective use of local generation and the time-of-day tariff while preventing peaks in aggregate demand at tariff boundaries by randomizing dispatch of loads.

This signaling and optimization methodology is fully described in [25,26] and the peaking risk that is mitigated, which has been identified in many simulation studies, is identified for example in [27,28].

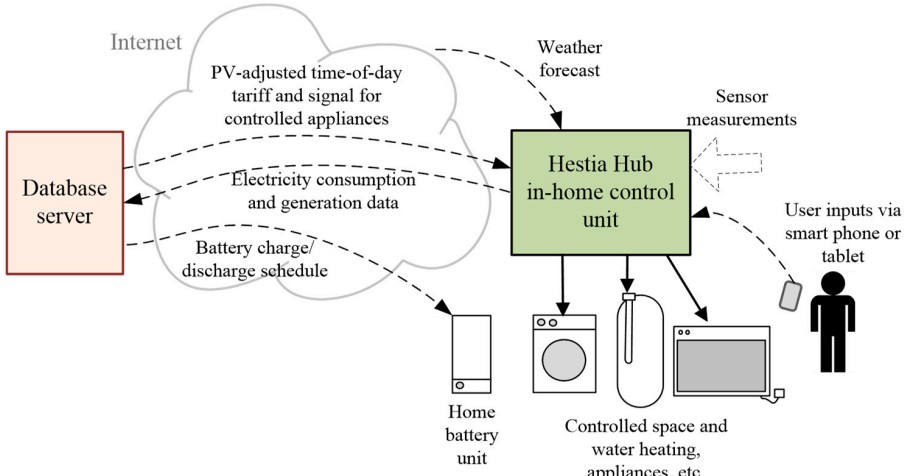

**Figure 2.** CEGADS system diagram.

An example page from the Hestia Hub display is shown in Figure 3a. All of the participants were given a smart plug as shown in Figure 3b for which the on/off status could be radio-controlled via a user interface provided by the Hestia unit. This allowed users to set a time window within which an appliance powered via the smart plug should operate, and the required operating duration. If some scheduling flexibility was available from the difference between the time window and the operating duration, the Hestia selected an optimized dispatch time using the demand response signal.

Nine of the participant households were equipped with a 2 kWh home battery unit (Figure 3c). The nine were deliberately chosen to exclude households with PV panels or thermal storage. These lithium-ion batteries were controlled to charge during low tariff periods and discharge during the early evening high tariff rate period, with the objective of improving the benefit these households obtained from the tariff scheme.

To summarise, the trial involved expanding the demand response potential in three villages by extending people's ideas of what community energy could do for them, by developing new activity in relation to the cooperative business model and tariff, and introducing technology in the shape of the Hestia control units, database server, smart plugs, batteries and display capability. In doing this, it was building on trust and knowledge that had been established through everyday social interactions and (for some) involvement in a local energy cooperative.

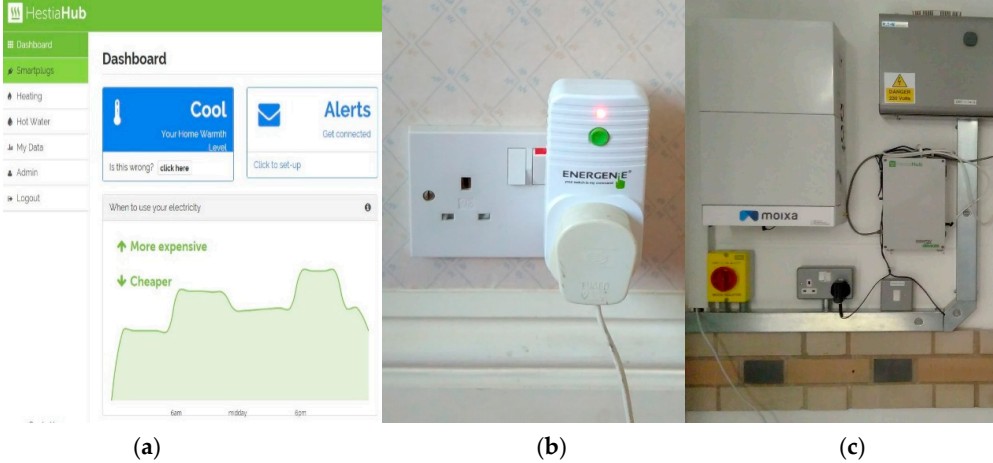

(**a**)  (**b**)  (**c**)

**Figure 3.** (**a**) Hestia Hub display. (**b**) Smart plug. (**c**) Home battery and heating control units.

## 3. Results from the Trial

### 3.1. Utilisation of Local Generation

Over the year of trial operation, out of the total PV generation of 43,406 kWh from the 14 generators, 18,307 kWh were used within the generating households, and 25,908 kWh were available to share with other participants. Of this available total, 22,154 kWh were matched with consumption using the algorithm described earlier, and the balance of 2944 kWh was allocated to the PPA. Figure 4 illustrates this outcome on a monthly basis. The generation tariffs gave an improved financial return simulated through the credit vouchers of about 80% to generators (£719 in addition to £868 from 50% deemed export feed-in tariff at £0.040/kWh making a total of £1587). The generation matched with consumption represented about 9.5% of the total electricity consumed (c. 233 MWh) by all the participants during the year.

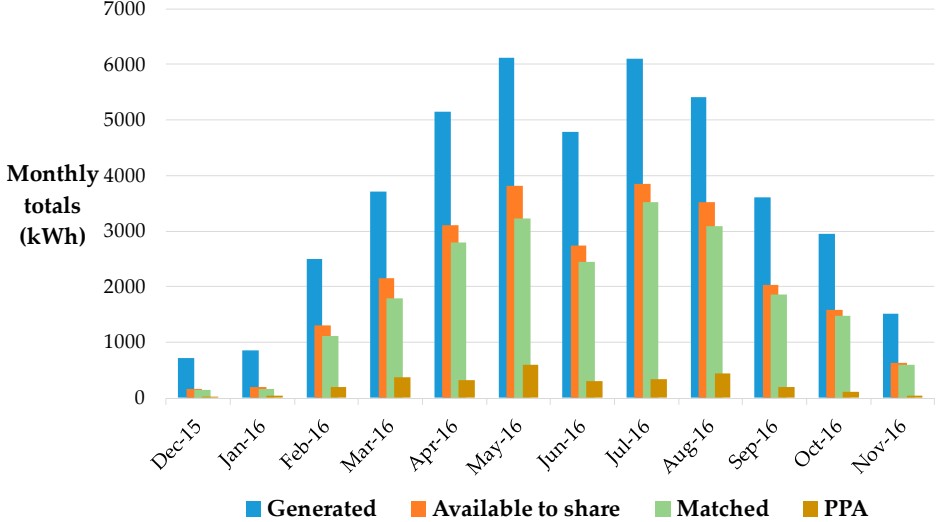

**Figure 4.** Allocation of PV generation by month.

The savings with respect to their existing tariffs that accrued to all participants are shown in Figure 5. The "tariff" plot shows the savings from the baseline time-of-day tariff. "Local use" shows the savings to participants who consumed the matched generation shown in Figure 4. "PV shared" shows the additional return to the PV generators.

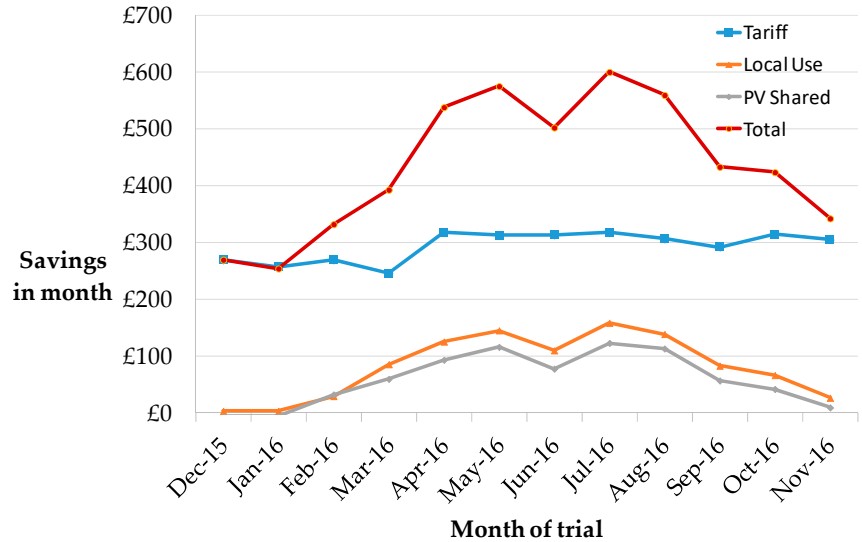

**Figure 5.** Financial savings to participants from the project.

### 3.2. Demand Side Response

The overall demand side response is illustrated in Figure 6a–d by comparing aggregate demand profiles at the start and end of the trial. The response derived from four sources:

- automatic "smart" control of water and space heating in six electrically-heated homes;
- time-shifting of supply via the home batteries installed in nine homes;
- semi-automatic time-shifting of demand using the smart plugs in all homes;
- decisions made by any of the participants to manually control any appliance taking account of the incentives offered.

The comparison between average consumption profiles in Figure 6 is influenced by the fact that December 2016 was much colder (295 degree-days) than December 2015 (154 degree-days). The six electrically-heated homes presented a special case in that they were already using a time-dependent tariff known as Economy 7. This comprises a low rate for 7 h overnight of about £0.07/kWh and a higher day rate of about £0.016 typically used, as in the present case, with thermal storage heating and domestic hot water tanks that can be charged at the low rate. So the automatic controls were given a signal which moved some of the heating demand into the middle of the day to take advantage of the local generation and lower mid-day tariff. The controls also ensured a more precise matching of stored thermal energy to the weather-dependent heating demand.

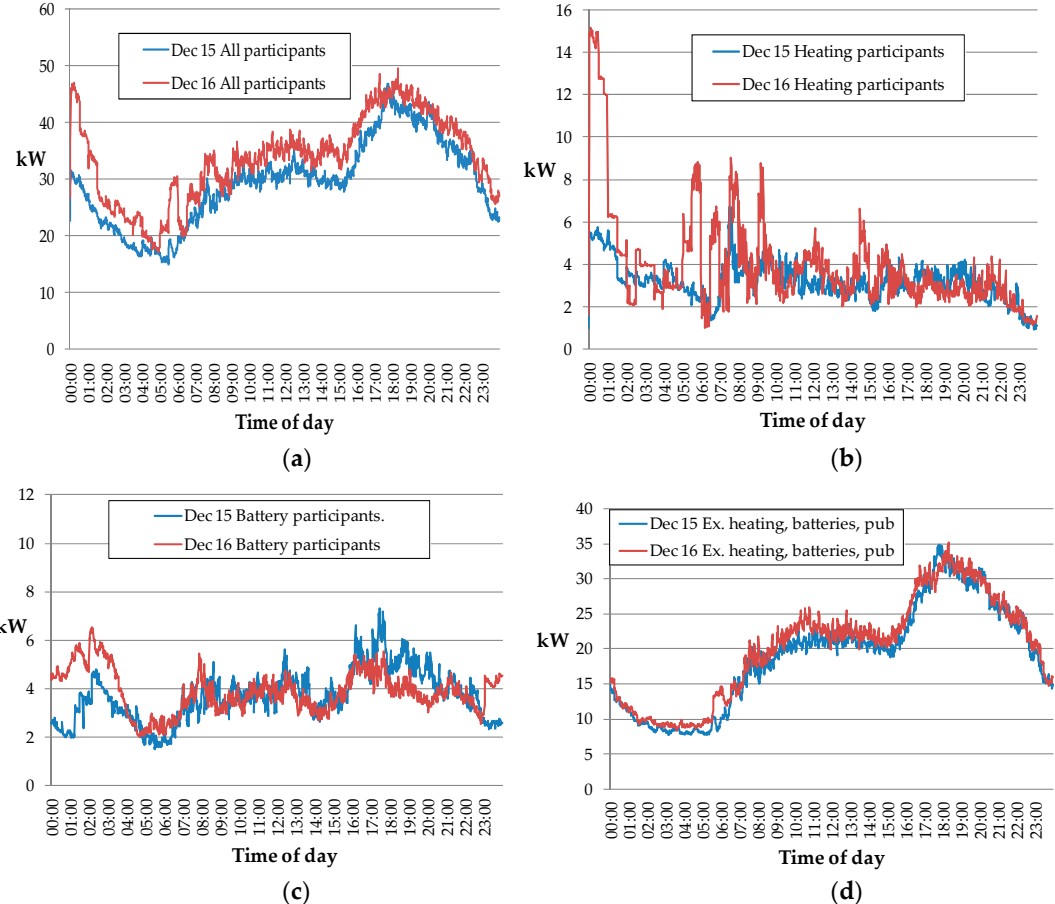

**Figure 6.** Comparison of average daily aggregate electricity use profile in December 2015 with December 2016, for different participant groups: (**a**) All participants (**b**) Participants with electric heating (**c**) Participants with batteries (**d**) All participants excluding those with batteries, electric heating and the pub.

The resulting shift in distribution of demand is illustrated in Figure 6b which shows the increased heating demand during the day and also a reduction of 16% in consumption during the peak tariff hours despite the colder weather in 2016. A comprehensive report focused on the performance of the heating controls is provided in [25].

The operation of eight batteries (one had to be removed before December 16) can be seen in Figure 6c with charging commencing at the start of the low tariff rate at 23:00 and continuing overnight. The reduction in demand by 20% during the evening high tariff period is also evident. It was found desirable to configure the batteries with a maximum discharge rate of 0.25 kW to ensure that the battery output always offset local consumption and was not exported, for which no reward could be offered. One of the participants with a battery also acquired an electric car during the trial year which contributed to the increased overnight demand seen in Figure 6c.

The relatively limited aggregate impact of the smart plugs and manual time-shifting can be seen in Figure 6d, which excludes the participants shown in Figure 6b,c and also the village pub (i.e., bar) which had a significant increase in power consumption during the year for commercial reasons unrelated to this trial. The participants in 6d had gas central heating so consumption was not greatly affected by the weather difference in the comparison. The increased overnight and mid-day consumption can be seen and also a slight reduction during the peak tariff time. The peak before 06:00 is believed to be wet appliance operation scheduled at the end of the low tariff period.

To examine the range of individual household responses, the average demand in each of the six tariff periods shown in Figure 1 was calculated for each household for October–December 2015 and for the same period in 2016. The correlation between changes in demand in each tariff period, and the tariff rate was then tested, with the hypothesis that demand would have changed over the year in inverse proportion to the tariff as consumers became accustomed to a time-of-day tariff and adjusted their demand accordingly. The results for different participant groups are shown in Table 1. A participant was counted as a "responder" in the table if a negative correlation was observed between change in demand and tariff rate with an $R^2$ value greater than 0.1. The much greater proportion of responders among participants with some additional technology that reinforces their engagement is evident. Note all the groups in Table 1 are independent i.e., there is no overlap of membership.

**Table 1.** Correlation of change in demand over a year with time-of-day tariff.

| Group Attribute | Number in Group | Responders | $R^2$ Range for Responders | Responders as % of Group |
|---|---|---|---|---|
| Controlled electric heating | 6 | 5 | 0.15–0.26 | 83% |
| PV generator | 14 | 11 | 0.13–0.7 | 79% |
| Battery storage | 8 | 7 | 0.23–0.82 | 88% |
| All other participants | 19 | 4 | 0.16–0.67 | 21% |

### 3.3. User Experience

Most participants in the CEGADS project were interviewed in three rounds of surveys; (by telephone, and online using Survey Monkey) around the start in late 2015 early 2016, during summer 2016, and spring 2017. They began with largely positive attitudes, with responses to questions concerning demand response and time of use tariffs as shown in Figure 7a,b. In round 1 of the interviews, most participants were positive about being able to switch the time of use of some of their electricity although the extent to which they could was determined by patterns of household occupation (i.e., if they were in the home during the daytime). Most participants felt able to switch devices such as white goods such as washing machines, tumble driers, and dishwashers and battery charging (e.g., for computers) and where time and space flexibility permitted (e.g., either being able to use them, or program them to run, in off peak hours).

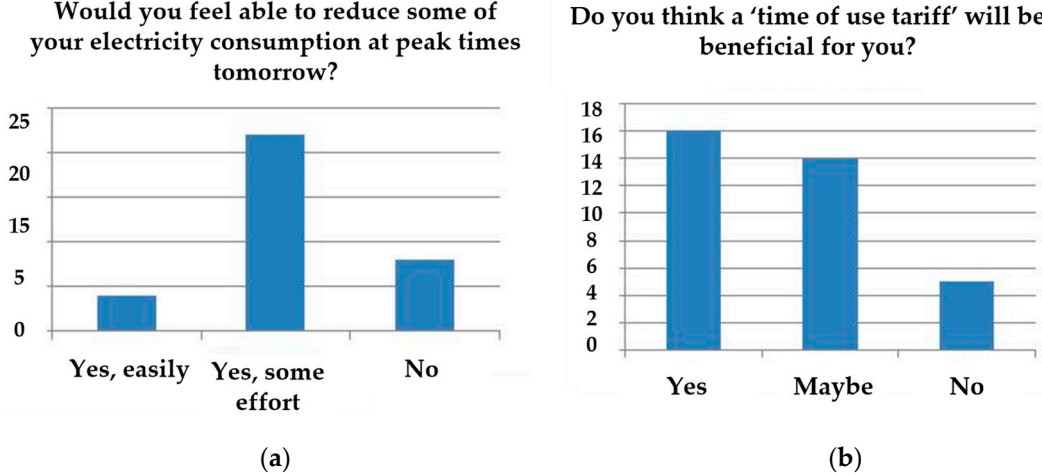

**Figure 7.** Initial perceptions of (**a**) demand response possibility and (**b**) a time-of-use tariff.

By the second round of interviews practices relating to the Hestia Hub technology provided were evident, with a range of usage levels as shown in Figure 8, and comments such as "*when I first started out I was looking at it daily or more frequently, but now . . . I'm more familiar with it*". Overall, 13 of the 37 (35%) respondents reported looking at their Hestia Hub at least once a week. This is roughly consistent with findings relating to in-home display usage by smart-metered customers in Great Britain as a whole, where 44% of householders reported that they were consulting their display at least once a week, between seven and 29 months after installation [20].

However, on the broad question as to whether the project had influenced day to day habits of electricity use, 31 (out of 39) said "*yes*". Of those who responded "*yes*", many reported shifting of activities, such as using the washing machine and dish washer at different times, or changing cooking practices, for example: "*sometimes opting to microwave or grill instead of using oven*". Of the 6 who responded no, some were already producing electricity through solar PV, some considered they had already made changes in their consumption, and some had family routines which they didn't want to shift.

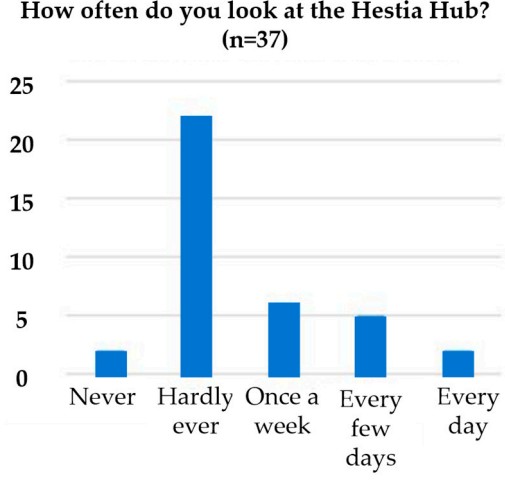

**Figure 8.** Engagement with smart metering and control device after six months.

The final survey revealed many practical changes in household practices that participants had taken, such as "*Weather—I was advised if I could use my washing machine when the sun was at its highest, that was the best time to do it*"; and "*Put the electric towel rail on a timer plug socket. It only operates sporadically in the peaks*". The motivation for demand response decisions was drawn from a wide range of factors as indicated in Figure 9, showing that the different channels used by the project

to communicate with participants all had a role in the results obtained. The variety of integrated approaches enabled participants to learn and incorporate their learning into new routines in different ways, through different means, and at different times, for example: *"when you have the reports that came through and here you have the cheque for your money and your report on your energy use, you could practically see how it all fitted together"*. Of 21 participants who said they had shifted usage during the trial, 17 said they would continue to do so, while 16 respondents who had reduced the amount of electricity usage during the trial were planning to maintain their reduction.

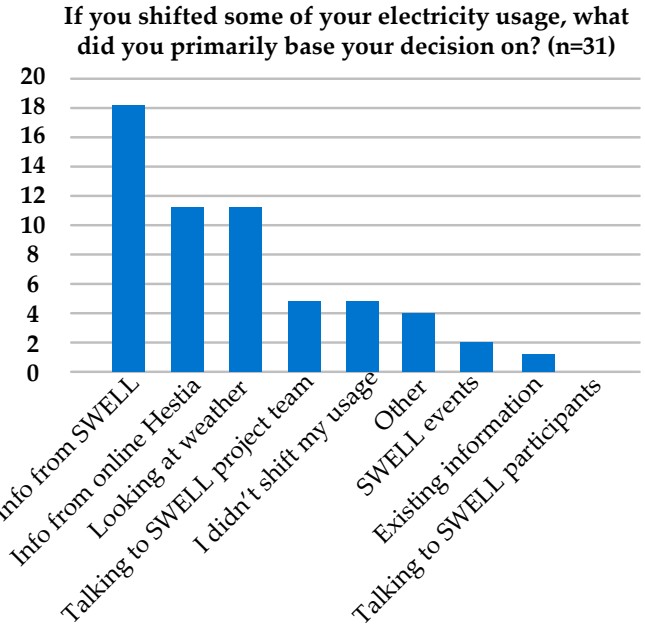

**Figure 9.** Decision basis for demand response actions.

## 4. Follow-On System Modelling and Commercial Implementation

### 4.1. Agent-Based Modelling

To investigate the wider applicability of this business and technical approach to localisation of energy use, an agent-based model (ABM) was constructed embodying both the technical features and the economic and social incentives of the real-life CEGADS trial. The model used the CASCADE framework [29] in which each agent is a household for which their energy-using appliances, energy flows, costs, and decisions are simulated taking account of their physical and social environment. The environmental factors included weather, tariffs, and the feedback and encouragement provided by participation in the project. The governing attributes of each agent (such as the number of household occupants and size of dwelling) were given values selected randomly from an appropriate range and distribution. Each run of the model (typically simulating a year's operation) therefore gave different outcomes, and as is conventional for an ABM, interpretation of results is based on the range of outcomes.

The initial correspondence between empirical and modelled results indicated that the model was useful. A series of tests were then undertaken to test how robust the positive results of the empirical trial were to changes in the scenario. Firstly, different weather files were used to investigate how dependent the overall results and benefits to individuals were on weather. This showed that the important characteristic of no participant being expected to lose out financially was preserved in differing weather conditions. Next, several communities were programmed to co-exist in the model and the smart signals sent to consumers in each model were examined after the model had evolved. It was noted that the signals exhibited similar characteristics, but that they evolved in a way that was specific to the community—indicating that the model was transferrable, but that the specific evolution

in response to the demographic and technology mix within the community would differ, resulting in different levels and patterns of demand response.

*4.2. Commercial Implementation*

The success of the CEGADS trial has led to a first fully commercial implementation of the concept for a community in the small town of Bethesda, North Wales [30]. This is based around a 100 kW micro hydro generator. 100 consumers have been recruited, who pay £0.07/kWh for matched use of local generation, and a small charge for membership of the co-operative club. Any electricity not supplied by the hydro is charged according to a time-of-day tariff similar to that in Figure 1. Because the output of this generator varies seasonally, power availability is signalled to users. Figure 10 shows the proportion of each user's power matched to low cost local generation, with an average of 65% over the year of operation. The overall shape of the graph reflects the availability of hydro generation, including the impact in May 2017 of a period of dry weather.

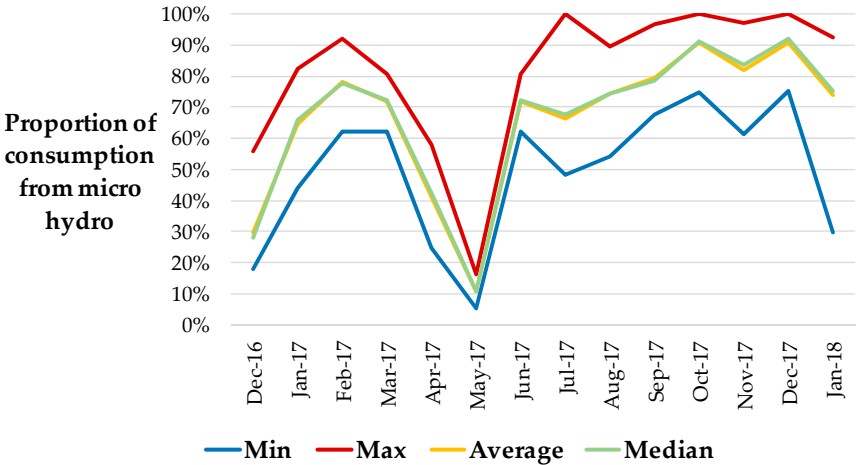

**Figure 10.** User demand matched to local micro-hydro.

The social enterprise promoting this model, Energy Local, is now developing a "starter pack" of processes and documentation [31] with follow-up support, allowing social enterprise clubs to be formed and the model implemented wherever appropriate generation, network configuration and community enthusiasm exist. New micro-hydro clubs are in progress elsewhere in Wales, and PV-based clubs in London and Gloucester.

**5. Discussion**

A limitation of this study is that there is no unambiguous counterfactual against which the impact on electricity use of the full range of interventions can be assessed. The process of recruiting participants and installing equipment in their homes inevitably involved engaging them with the objectives of the study. So the comparison of electricity demand measurements from the end of the study with those at the start cannot fully demonstrate the behavioural shifts that may have occurred, but they do reliably show the effect of the heating control and battery technologies. With further development, the ABM modelling techniques tested as part of this study could provide policymakers with additional predictive insight on the impact of wide-scale adoption of this kind of energy localisation scheme.

The increased demand response from those participants who had a significant supporting technology is very evident from Table 1. The smart plugs that were given to all participants clearly did not have the same impact as the other technologies. The increased response from participants with controlled electric heating and PV is notable since both started the trial with an incentive to attend to the timing of electricity demand, in the heating case through their longstanding use of the Economy 7 tariff, and in the PV case to make use of their own generation. So it must be concluded that their initial

sensitivity to timing was further reinforced through the various inputs categorized in Figure 7, and for the heating users by the control signal alignment with the tariff. For the battery-equipped participants, the response seen fully reflects the automatic operation of the technology because the batteries were installed in January 2016. For the remaining participants it is evident from the data that only a few enthusiasts were able to sustain an increasing level of demand response that could be detected in the start-to-finish comparison. This does not of course preclude that many of this group may have taken demand response actions on an intermittent basis as suggested by the interviews. These results are very consistent with previous studies such as the review of 21 trials by Frontier Economics [32] which has a key finding "Interventions to automate responses deliver the greatest and most sustained household shifts in demand where consumers have certain flexible loads".

The financial benefit of the business model is clear from Figure 5, amounting to an average of £109 to each participant over a year, on average consumption of 4854 kWh that would cost £688 at the benchmark fixed rate used as a comparator of £0.135/kWh and standing charge of £0.09/day. These substantial savings illustrate the value that will be made accessible by the UK national smart meter rollout as long as regulation, and suitable smart meter data processing capability, facilitates this form of community energy scheme and tariff structure. The way in which the benefit of the PV generation is spread across the community makes this model particularly attractive for social housing, where otherwise the restriction of local generation to a subset of dwellings with favourable roof orientation can lead to perception of unfairness [33].

The recent UK regulatory review [22] has identified the issue that consumers who benefit from local generation, as in the present scheme, pay less towards the distribution and transmission infrastructure through per-kWh tariffs, but the burden they place on that infrastructure is determined by their peak network demand. This is ultimately likely to lead to tariff structures that are less dependent on the volume of electricity consumed and have some dependency on peak demand, such as those proposed by Simshauser [34] and Nijhuis et al. [35]. The battery and smart heating control technologies demonstrated in this project directly address this issue by substantially reducing peak demand.

This trial also provides a more positive perspective on the economics of home batteries than other trials such as Uddin et al. [36] who tested exactly the same product as used in the present project and concluded there was no saving from increased self-consumption of PV generation and a substantial loss to the user from battery depreciation. It is worth noting that the trial reported by Uddin et al. was purely techno-economic, with empirical testing in a single household as the basis for economic modelling: there was no community or social dimension. In the case of the CEGADS/SWELL trial, the battery users made savings at an average rate of about £23 per annum from tariff arbitrage, by avoiding 1 kWh per day in the evening peak rate period and drawing the corresponding charge overnight. Clearly this alone would not justify the investment, or cover the depreciation, but in combination with other grid services such as Short Term Operating Reserve, for which the UK system operator National Grid currently pays about 12 p/kWh [37] adding c. £10 per annum to the return, a path to viability, as battery costs fall, can be seen. Another role for batteries relevant to community energy is to facilitate connection of generation capacity. In [38] Idlbi et al. elaborate a case study in which 0.5 MWh of lithium battery capacity enables connection of 3 MWp PV generation to a network that would otherwise require reinforcement at much greater cost to maintain voltage compliance. However, batteries are not essential to the value of the Energy Local model, as shown by the results from the Bethesda follow-up project, which did not deploy them.

## 6. Conclusions

This trial, combined with the subsequent commercial implementation of the business model, has demonstrated that valuable technical, economic, and social outcomes can be achieved by localisation of electricity generation and consumption within a community-of-place-based organisational framework. The generally positive user experience described in Section 3 reflects

the support given to participants by the project team allowing them to understand and engage with the novel technology and tariffs. The business model delivered useful financial savings for consumers which have been shown to be repeatable elsewhere, while the technology successfully demonstrated the use of both electrical and thermal energy storage to reshape the daily profile of electricity demand, in response to technical and financial signals, such that peak demand is reduced and local consumption of local generation is increased. This demand response was stronger for the automated mechanisms managing energy storage than from the simpler devices providing appliance scheduling which required repeated user configuration. These results show that the sustainability of the transition to renewable energy can be strengthened with a community-oriented approach that supports users through technological change and improves the return on investment by localising generation and consumption. The enterprises and institutions taking part in this trial are now seeking to build on this experience by evolving the business model and technology so that they can be widely deployed, and are also motivating adjustments to the regulatory environment that facilitate local initiatives and consumer engagement.

**Author Contributions:** Formal analysis, P.B., J.R.S., R.M. and J.H.; Investigation, P.B., J.R.S., R.M., J.H. and S.D.; Writing—original draft, P.B., J.R.S., R.M., J.H. and S.D.

**Funding:** As detailed under Acknowledgements and Conflicts of Interest below.

**Acknowledgments:** The authors would like to thank the Engineering and Physical Sciences Research Council (EPSRC) and Innovate UK for providing the financial support for this study as part of the CEGADS project (EP/M507209/1 and EP/M507210/1), including funds for covering the costs to publish in open access. The project implementation was also part funded by Energy Local (Development) Ltd., Exergy Devices Ltd., Moixa Technology Ltd., Westmill Sustainable Energy Trust, and Co-Operative Energy who provided the supermarket vouchers to participants reflecting their benefit from the simulated tariffs.

**Conflicts of Interest:** Two authors (Boait and Morris) have roles in the enterprises acknowledged above that provided match funding and technology for the project. None of the other authors has a conflict of interest which could inappropriately influence this work. The results reported were generated and reviewed by all the authors. This work (i.e., the data analysis and drafting of the manuscript) was funded by the UK Engineering and Physical Sciences Research Council and Innovate UK. They had no role in the collection of data or in its analysis and interpretation in the paper and none in the decision to submit to this journal.

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
