# Peer review of "The Practice and Potential of Renewable Energy Localisation: Results from a UK Field Trial"

_sustainability, doi:10.3390/su11010215_

Round 1

Reviewer 1 Report

The manuscript is interesting but requires extra efforts to improve its quality and presentation for the prestigious journal Sustainability. A set of comments are expounded hereafter.

- The manuscript is, in general, well written. However, there are some mistakes or improvements regarding the format of the document, as commented below.

The “A” in the title should not be capitalized.

In line 152, there is no reason to put “smart” between quotations. The same occurs for “smart plug” in line 174. It must be noted that this is only a humble opinion.

In line 159, a coma should be placed before “metering”.

The citation of subfigures of the figure 3 is not adequate (lines 174 and 180). The same occurs for figures 4, 5, etc.

The caption of figure 3 lacks the terminal period (punctuation).

This reviewer suggests improving the aspect of the figures by removing the frame. For instance, figures 8 and 9 already do not include such frame.

The format of the references must be revised according to the template of the Journal. For instance, the abbreviated name of the journals must be used.

In line 531, the year is not adequate; 2017 should be replaced by 2019. The same occurs for the page headers.

- About the content of the manuscript, it covers a very interesting topic. The comments after a careful revision are the following:

In the Keywords, in this reviewer opinion, “renewable energy” could be added if the authors agree with the proposal. Indeed, “business model” could also be considered.

The contextualization of the research is properly scheduled. Nonetheless, a concept to mention is “smart home” given the fact that the equipment that has been applied provides certain smart features to the households.

In line 106, the goals of a project are mentioned. However, there has not been previous indication of such a project; only a trial has been commented. Therefore, for a clear presentation, this reviewer suggests solving this issue by explicitly mentioning the CEGADS project/trial in this paragraph.

Concerning the name of the project (or trial), in the second section it is said that CEGADS and SWELL are both used. Along the manuscript, the trial is sometimes named as CEGADS and other times, as SWELL. For a more coherent presentation, this reviewer recommends using only one of both acronyms for the whole paper, if the authors agree with the suggestion.

A brief sentence to explain in a concise manner the structure and/or content of the section is desirable when various subsections are found. This is commented for sections 2, 3 and 4.

In lines 125-126, the installation of metering and monitoring systems in each household is commented. But, did this equipment involve some cost for the user? This issue should be mentioned in a brief manner.

In Figure 2, it is seen that the user accesses the acquired data through smartphone or tablet. However, this relevant issue is not described in the text, which should be done. In addition, the web-based access should also be explicitly indicated due to the fact that it appears among the features of the trial expounded in the Introduction. Even more, if possible, a screenshot of the web interface would facilitate the understanding of the management experience that the users have experienced.

Where is the database server located? Does this database belong to the Hestia products? Is the battery (line 180) of lead-acid type or Lithium-ion? These details can be useful for the interested reader.

A flowchart or table to illustrate the implemented control strategy would help to better visualize the energy flows between the involved components (PV panels, battery, appliances, etc.).

In line 281, the “Hub technology” is commented regarding the users opinions; however, this term has not been previously introduced. This reviewer understands that it is referred to the metering and monitoring system based on the Hestia equipment and the availability of accessible data. For a clear presentation to the reader, this concept or term should be indicated before, perhaps, it could be done in the subsection 2.2. Indeed, the Hestia Hub appears in figure 2.

This reviewer would like to remark the valuable contribution that papers reporting experimental results provide. This is, therefore, a positive aspect of the manuscript.

Given the scope of the Journal, sustainability or sustainable development should be mentioned, at least in a brief manner, in the Discussion or in the Conclusions.

Some future guidelines that the authors are considering on the view of the presented work would enrich the Conclusions section.

As a conclusion of the revision, if all the described suggestions are addressed, the manuscript will reach a better presentation and scientific level, according to the prestigious journal Sustainability.

Author Response

Please see the attached PDF file covering both reviewers.

Reviewer 2 Report

The manuscript emphasizes the importance of renewable energies localization by collecting data from 48 homes in the UK. A systematic study was designed and followed and the results were well presented. The manuscript is well written, and it provides sufficient novel results, which I think make it qualified to be published in Sustainability journal. I just have two minor revisions:

 - In the abstract, please briefly mention the main achievements. The current version mostly covers the methods rather than the results.

 - The graphs need to be better labeled and formatted. For examples, the fonts and colors must be consistent. 

Author Response

Please see the attached PDF responding to both reviewers.

Round 2

Reviewer 1 Report

The efforts carried out by the authors are appreciable in the revised manuscript. The provided suggestions have been addressed in a rapid and proper way.

Therefore, after a careful revision, this reviewer considers that the revised manuscript has been noticeably enhanced and is suitable for publication in Sustainability in present form.